# A Bio-inspired Redundant Sensing Architecture

**Anh Tuan Nguyen, Jian Xu and Zhi Yang**[*]
Department of Biomedical Engineering
University of Minnesota
Minneapolis, MN 55455
[*]`yang5029@umn.edu`

## Abstract

Sensing is the process of deriving signals from the environment that allows artificial systems to interact with the physical world. The Shannon theorem specifies the maximum rate at which information can be acquired [1]. However, this upper bound is hard to achieve in many man-made systems. The biological visual systems, on the other hand, have highly efficient signal representation and processing mechanisms that allow precise sensing. In this work, we argue that redundancy is one of the critical characteristics for such superior performance. We show architectural advantages by utilizing redundant sensing, including correction of mismatch error and significant precision enhancement. For a proof-of-concept demonstration, we have designed a heuristic-based analog-to-digital converter - a zero-dimensional quantizer. Through Monte Carlo simulation with the error probabilistic distribution as a priori, the performance approaching the Shannon limit is feasible. In actual measurements without knowing the error distribution, we observe at least 2-bit extra precision. The results may also help explain biological processes including the dominance of binocular vision, the functional roles of the fixational eye movements, and the structural mechanisms allowing hyperacuity.

## 1   Introduction

Visual systems have perfected the art of sensing through billions of years of evolution. As an example, with roughly 100 million photoreceptors absorbing light and 1.5 million retinal ganglion cells transmitting information [2, 3, 4], a human can see images in three-dimensional space with great details and unparalleled resolution. Anatomical studies determine the spatial density of the photoreceptors on the retina, which limits the peak foveal angular resolution to 20-30 arcseconds according to Shannon theory [1, 2]. There are also other imperfections due to nonuniform distribution of cells' shape, size, location, and sensitivity that further constrain the precision. However, experiment data have shown that human can achieve an angular separation close to 1 arcminute in a two-point acuity test [5]. In certain conditions, it is even possible to detect an angular misalignment of only 2-5 arcseconds [6], which surpasses the virtually impossible physical barrier. This ability, known as *hyperacuity*, has baffled scientists for decades: what kind of mechanism allows human to read an undistorted image with such a blunt instrument?

Among the approaches to explain this astonishing feat of human vision, redundant sensing is a promising candidate. It is well-known that redundancy is an important characteristic of many biological systems, from DNA coding to neural network [7]. Previous studies [8, 9] suggest there is a connection between hyperacuity and *binocular vision* - the ability to see images using two eyes with overlapping field of vision. Also known as *stereopsis*, it presents a passive form of redundant sensing. In addition to the obvious advantage of seeing objects in three-dimensional space, the binocular vision has been proven to increase visual dynamic range, contrast, and signal-to-noise ratio [10]. It is evident that seeing with two eyes enables us to sense a higher level of information

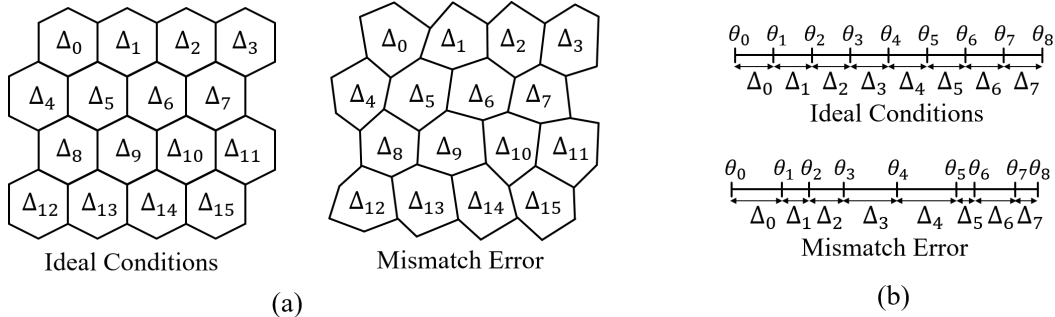

Figure 1: Illustration of $n$-dimensional quantizers without (ideal) and with mismatch error. (a) Two-dimensional quantizers for image sensing. (b) Zero-dimensional quantizers for analog-to-digital data conversion.

as well as to correct many intrinsic errors and imperfections. Furthermore, the eyes continuously and involuntarily engage in a complex micro-fixational movement known as *microsaccade*, which suggests an active form of redundant sensing [11]. During microsaccade, the image projected on the retina is shifted across a few photoreceptors in a pseudo-random manner. Empirical studies [12] and computational models [13] suggest that the redundancy created by these micro-movements allows efficient sampling of spatial information that can surpass the static diffraction limitation.

Both biological and artificial systems encounter similar challenges to achieve precise sensing in the presence of non-ideal imperfections. One of those is *mismatch error*. At a high resolution, even a small degree of mismatch error can degrade the performance of many man-made sensors [14, 15]. For example, it is not uncommon for a 24-bit analog-to-digital converter (ADC) to have 18-20 bits effective resolution [16]. Inspired by the human visual system, we explore a new computational framework to remedy mismatch error based on the principle of redundant sensing. The proposed mechanism resembles the visual systems' binocular architecture and is designed to increase the precision of a zero-dimensional data quantization process. By assuming the error probabilistic distribution as a priori, we show that precise data conversion approaching the Shannon limit can be accomplished.

As a proof-of-concept demonstration, we have designed and validated a high-resolution ADC integrated circuit. The device utilizes a heuristic approach that allows unsupervised estimation and calibration of mismatch error. Simulation and measurement results have demonstrated the efficacy of the proposed technique, which can increase the effective resolution by 2-5 bits and linearity by 4-6 times without penalties in chip area and power consumption.

## 2 Mismatch Error

### 2.1 Quantization & Shannon Limit

Data quantization is the partition of a continuous $n$-dimensional vector space into $M$ subspaces, $\Delta_0, ..., \Delta_{M-1}$, called *quantization regions* as illustrated in Figure 1. For example, an eye is a two-dimensional biological quantizer while an ADC is a zero-dimensional artificial quantizer, where the partition occurs in a spatial, temporal and scalar domain. Each quantization region is assigned a representative value, $d_0, ..., d_{M-1}$, which uniquely encodes the quantized information. While the representative values are well-defined in the abstract domain, the actual partition often depends on the physical properties of the quantization device and has a limited degree of freedom for adjustment. An optimal data conversation is achieved with a set of uniformly distributed quantization regions. In practice, it is difficult to achieve due to the physical constraints in the partition process. For example, individual pixel cells can deviate from the ideal morphology, location, and sensitivity. These relative differences, referred to as *mismatch error*, contribute to the data conversion error.

In this paper, we consider a zero-dimensional (scalar) quantizer, which is the mathematical equivalence of an ADC device. A $N$-bit quantizer divides the continuous conversion full-range (FR = $[0, 2^N]$) into $2^N$ quantization regions, $\Delta_0, ..., \Delta_{2^N-1}$, with nominal unity length $E(|\Delta_i|) = \Delta = 1$

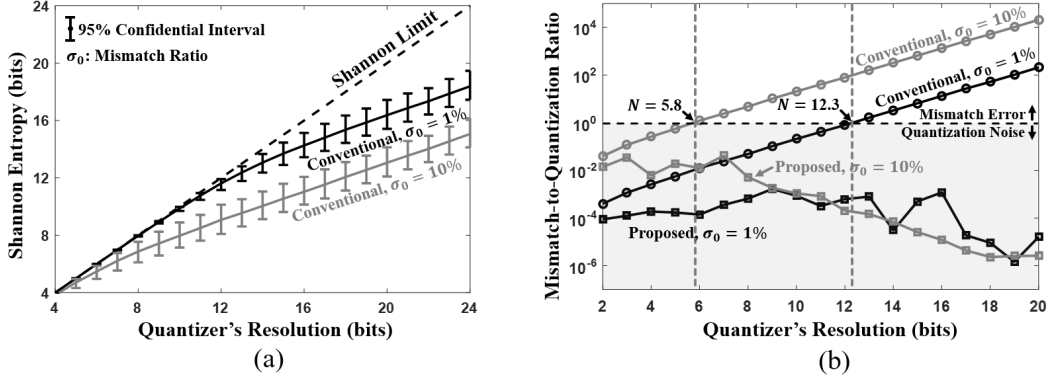

Figure 2: (a) Degeneration of entropy, i.e. maximum effective resolution, due to mismatch error versus quantizer's intrinsic resolution. (b) The proportion of data conversion error measured by mismatch-to-quantization ratio (MQR). With a conventional architecture, mismatch error is the dominant source, especially in a high-resolution domain. The proposed method allows suppressing mismatch error below quantization noise and approaching the Shannon limit.

least-significant-bit (LSB). The quantization regions are defined by a set of discrete references[1], $S_R = \{\theta_0, ..., \theta_{2^N}\}$, where $0 = \theta_0 < \theta_1 < ... < \theta_{2^N} = 2^N$. An input signal $x$ is assigned the digital code $d(x) = i \in S_D = \{0, 1, 2, ..., 2^N - 1\}$, if it falls into region $\Delta_i$ defined by

$$x \leftarrow d(x) = i \quad \Leftrightarrow \quad x \in \Delta_i \quad \Leftrightarrow \quad \theta_i \leq x < \theta_{i+1}. \tag{1}$$

The Shannon entropy of a $N$-bit quantizer [17, 18] quantifies the maximum amount of information that can be acquired by the data conversion process

$$H = -\log_2 \sqrt{12 \cdot M}, \tag{2}$$

where $M$ is the normalized total mean square error integrated over each digital code

$$
\begin{aligned}
M &= \frac{1}{2^{3N}} \int_0^{2^N} [x - d(x) - 1/2]^2 dx \\
&= \frac{1}{2^{3N}} \sum_{i=0}^{2^N-1} \int_{\theta_i}^{\theta_{i+1}} (x - i - 1/2)^2 dx.
\end{aligned} \tag{3}
$$

In this work, we consider both quantization noise and mismatch error. The *Shannon limit* is generally preferred as the maximum rate at which information can be acquired without any mismatch error, where $\theta_i = i, \forall i$ or $S_R \backslash \{2^N\} = S_D$, $M$ is equal to the total quantization noise $Q = 2^{-2N}/12$, and the entropy is equal to the quantizer's intrinsic resolution $H = N$. The differences between $S_R \backslash \{2^N\}$ and $S_D$ are caused by mismatch error and result in the degeneration of entropy. Figure 2(a) shows the entropy, i.e. maximum effective resolution, versus the quantizer's intrinsic resolution with fixed mismatch ratios $\sigma_0 = 1\%$ and $\sigma_0 = 10\%$. Figure 2(b) describes the proportion of error contributed by each source, as measured by mismatch-to-quantization ratio (MQR)

$$\text{MQR} = \frac{M - Q}{Q}. \tag{4}$$

It is evident that at a high resolution, mismatch error is the dominant source causing data conversion error. The Shannon theory implies that mismatch error is the fundamental problem relating to the physical distribution of the reference set. [19, 20] have proposed post-conversion calibration methods, which are ineffective in removing mismatch error without altering the reference set itself. A standard workaround solution is using larger components thus better matching characteristics; however, this incurs penalties concerning cost and power consumption. As a rule of thumb, 1-bit increase in resolution requires a 4-time increase of resources [14]. To further advance the system performance, a design solution that is robust to mismatch error must be realized.

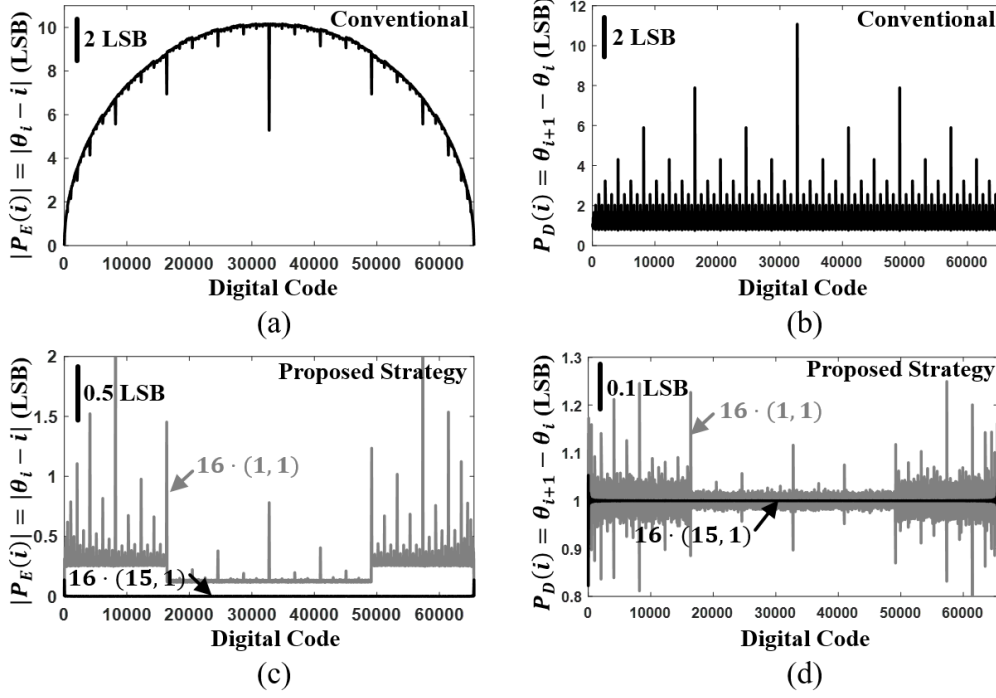

Figure 3: Simulated distribution of mismatch error in terms of (a) expected absolute error $|\mathcal{P}_E(i)|$ and (b) expected differential error $\mathcal{P}_D(i)$ in a 16-bit quantizer with 10% mismatch ratio. (c, d) Optimal mismatch error distribution in the proposed strategy. At the maximum redundancy $16 \cdot (15, 1)$, mismatch error becomes negligible.

## 2.2 Mismatch Error Model

For artificial systems, binary coding is popularly used to encode the reference set. It involves partitioning the array of unit cells into a set of binary-weighted components $S_C$, and assembling different components in $S_C$ to form the needed references. The precision of the data conversion is related to the precise matching of these unit cells, which can be in forms of comparators, capacitors, resistors, or transistors, etc. Due to fabrication variations, undesirable parasitics, and environmental interference, each unit cell follows a probabilistic distribution which is the basis of mismatch error. We consider the situation where the distribution of mismatch error is known as a priori. Each unit cell, $c_u$, is assumed to be normally distributed with mismatch ratio $\sigma_0$: $c_u \sim \mathcal{N}(1, \sigma_0^2)$. $S_C$ is then a collection of the binary-weighted components $c_i$, each has $2^i$ independent and identically distributed unit cells

$$S_C = \{c_i | c_i \sim \mathcal{N}(2^i, 2^i \sigma_0^2)\}, \quad \forall i \in [0, N-1]. \tag{5}$$

Each reference $\theta_i$ is associated with a unique assembly $X_i$ of the components[2]

$$S_R \backslash \{2^N\} = \{\theta_i = \frac{\sum_{c_k \in X_i} c_k}{\frac{1}{2^N - 1} \sum_{j=0}^{N-1} c_j} | X_i \in \mathcal{P}(S_C)\}, \quad \forall i \in [0, 2^N - 1], \tag{6}$$

where $\mathcal{P}(S_C)$ is the power set of $S_C$. Binary coding allows the shortest data length to encode the references: $N$ control signals are required to generate $2^N$ elements of $S_R$. However, because each reference is bijectively associated with an assembly of components, it is not possible to rectify the mismatch error due to the random distribution of the components' weight without physically altering the components themselves.

The error density function defined as $P_E(i) = \theta_i - i$ quantifies the mismatch error at each digital code. Figure 3(a) shows the distribution of $|P_E(i)|$ at 10% mismatch ratio through Monte Carlo

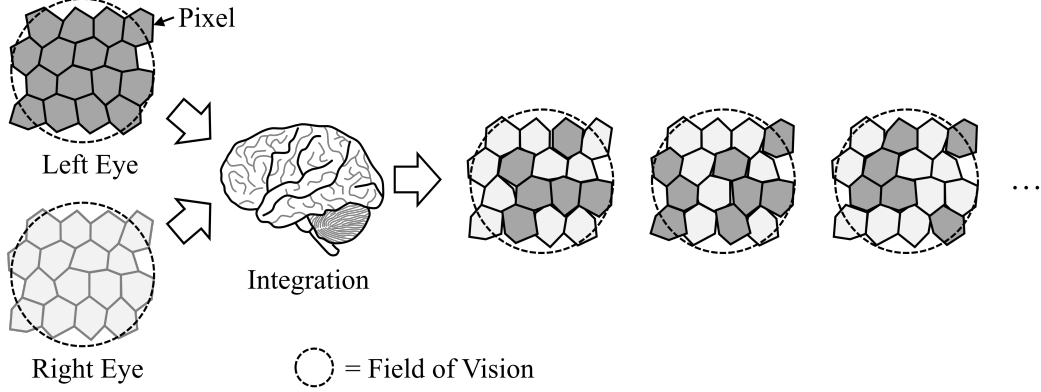

Figure 4: Associating and exchanging the information between individual pixels in the same field of vision generate an exponential number of combinations and allow efficient spatial data acquisition beyond physical constraints. Inspired by this process, we propose a redundant sensing strategy that involves blending components between two imperfect sets to gain extra precision.

simulations, where there is noticeably larger error associating with middle-range codes. In fact, it can be shown that if unit cells are independent, identically distributed, $P_E(i)$ approximates a normal distribution as follows

$$P_E(i) = \theta_i - i \sim \mathcal{N}(0, \sum_{j=0}^{N-1} 2^{j-1} \left| D_j - \frac{i}{2^N - 1} \right| \sigma_0^2), \quad i \in [0, 2^N - 1], \tag{7}$$

where $i = \overline{D_{N-1}...D_1 D_0}$ ($D_j \in \{0, 1\}, \forall j$) is the binary representation of $i$.

Another drawback of binary coding is that it can create differential "gap" between the references. Figure 3(b) presents the estimated distribution of differential gap $P_D(i) = \theta_{i+1} - \theta_i$ at 10% mismatch ratio. When the gap exceeds two unit-length, signals that should be mapped to two or multiple codes collapse into a single code, resulting in a loss of information. This phenomenon is commonly known as *wide code*, an unrecoverable situation by any post-conversion calibration methods. Also, wide gaps tend to appear at two adjacent codes that have large Hamming distance, e.g. $\overline{01111}$ and $\overline{10000}$. Subsequently, the amount of information loss can be signal dependent and amplified at certain parts of data conversation range.

## 3   Proposed Strategy

The proposed general strategy is to incorporate redundancy into the quantization process such that one reference $\theta_i$ can be generated by a large number of distinct component assemblies $X_i$, each yields a different amount of mismatch. Among numerous options that lead to the same goal, the optimal reference set is the collection of assemblies with the least mismatch error over every digital code.

Furthermore, we propose that such redundant characteristic can be achieved by resembling the visual systems' binocular structure. It involves a secondary component set that has overlapping weights with the primary component set. By exchanging the components with similar weights between the two sets, excessive redundant component assemblies can be realized. We hypothesize that a similar mechanism may have been employed in the brain that allows associating information between individual pixels on the same field of vision in each eye as illustrated in Figure 4. Because such association creates an exponential number of combinations, even a small percentage of 100 million photoreceptors and 1.5 million retinal ganglion cells that are "interchangeable" could result in a significant degree of redundancy.

The design of the primary and secondary component set, $S_{C,0}$ and $S_{C,1}$, specifies the level and distribution of redundancy. Specifically, $S_{C,1}$ is derived by subtracting from the conventional binary-weighted set $S_C$, while the remainders form the primary component set $S_{C,0}$. The total nominal weight remains unchanged as $\sum_{c_{i,j} \in (S_{C,0} \cup S_{C,1})} \overline{c}_{i,j} = 2^{N_0} - 1$, where $N_0$ is the resolution of the

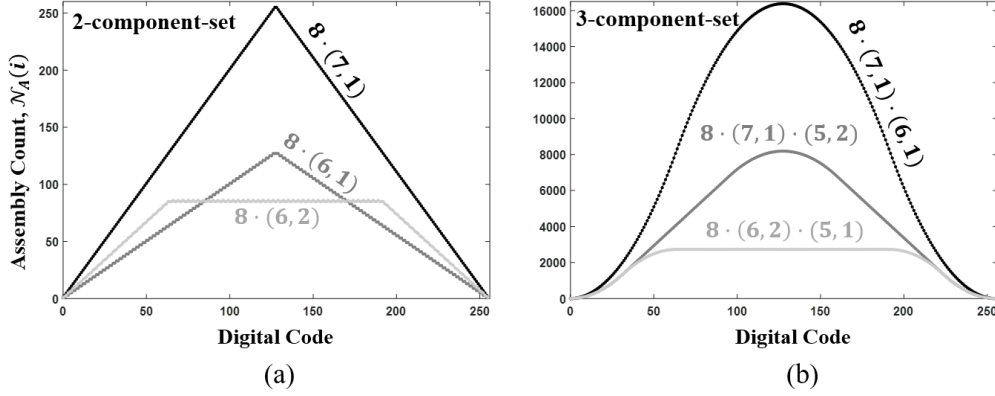

Figure 5: The distribution of the number of assemblies $\mathcal{N}_A(i)$ with different geometrical identity in (a) 2-component-set design and (b) 3-component-set design. Higher assembly count, i.e., larger level of redundancy, is allocated for digital codes with larger mismatch error.

quantizer as well as the primary component set. It is worth mentioning that mismatch error is mostly contributed by the most-significant-bit (MSB) rather than the least-significant-bit (LSB) as implied by Equation (5). Subsequently, to optimize the level and distribution of redundancy, the secondary set should advantageously consist of binary-weighted components that are derived from the MSB. $S_{C,0}$ and $S_{C,1}$ can be described as follows

$$\text{Primary: } S_{C,0} = \left\{ c_{0,i} \middle| \bar{c}_{0,i} = \begin{cases} 2^i, & \text{if } i < N_0 - N_1 \\ 2^i - \bar{c}_{1,i-N_0+N_1}, & \text{otherwise} \end{cases}, \forall i \in [0, N_0 - 1] \right\},$$

$$\text{Secondary: } S_{C,1} = \{ c_{1,i} | \bar{c}_{1,i} = 2^{N_0 - N_1 + i - s_1}, \forall i \in [0, N_1 - 1] \},$$

(8)

where $N_1$ is the resolution of $S_{C,1}$ and $s_1$ is a scaling factor satisfying $1 \le N_1 \le N_0 - 1$ and $1 \le s_1 \le N_0 - N_1$. Different values of $N_1$ and $s_1$ result in different degree and distribution of redundancy. Any design within this framework can be represented by its unique geometrical identity: $N_0 \cdot (N_1, s_1)$. The total number of components assemblies is $|\mathcal{P}(S_{C,0} \cup S_{C,1})| = 2^{N_0 + N_1}$, which is much greater than the cardinality of the reference-set $|S_R| = 2^{N_0}$, thus implies the high level of intrinsic redundancy.

$\mathcal{N}_A(i)$ is defined as the number of assemblies that represent the same reference $\theta_i$ and is an essential indicator that specifies the redundancy distribution

$$\mathcal{N}_A(i) = |\{ X | X \in \mathcal{P}(S_{C,0} \cup S_{C,1}) \wedge \sum_{c_{j,k} \in X} \bar{c}_{j,k} = i \}|, \quad i \in [0, 2^{N_0} - 1].$$

(9)

Figure 5(a) shows $\mathcal{N}_A(i)$ versus digital codes with $N_0 = 8$ and multiple combinations of $(N_1, s_1)$. The design of $S_{C,1}$ should generate more options for middle-range codes, which suffer from larger mismatch error. Simulations suggest $N_1$ decides the total number of assemblies, $\sum_{i=0}^{2^{N_0}-1} \mathcal{N}_A(i) = |\mathcal{P}(S_{C,0} \cup S_{C,1})| = 2^{N_0 + N_1}$; $s_1$ defines the morphology of the redundancy distribution. A larger value of $s_1$ gives a more spreading distribution.

Removing mismatch error is equivalent to searching for the optimal component assembly $X_{op,i}$ that generates the reference $\theta_i$ with the least amount of mismatch

$$X_{op,i} = \underset{X \in \mathcal{P}(S_{C,0} \cup S_{C,1})}{\operatorname{argmin}} \left| i - \sum_{c_{j,k} \in X} c_{j,k} \right|, \quad i \in [0, 2^{N_0} - 1].$$

(10)

The optimal reference set $S_{R,op}$ is then the collection of all references generated by $X_{op,i}$. In this work, we do not attempt to find $X_{op,i}$ as it is an NP-optimization problem with the complexity of $O(2^{N_0 + N_1})$ that may not have a solution in the polynomial space. Instead, this section focuses on showing the achievable precision with the proposed architecture while section 4 will describe a heuristic approach. The simulation results in Figure 2(b) demonstrate our technique can suppress

mismatch error below quantization noise, thus approaching the Shannon limit even at high resolution and large mismatch ratio. In this simulation, the secondary set is chosen as $N_1 = N_0 - 1$ for maximum redundancy. Figure 3(c, d) shows the distribution of mismatch error after correction. Even at the minimum redundancy ($N_1 = 1$), a significant degree of mismatch is rectified. At the maximum redundancy ($N_1 = N_0 - 1$), the mismatch error becomes negligible compared with quantization noise.

Based on the same principles, a $n$-set components design ($n = 3, 4, ...$) can be realized, which gives an increased level redundancy and more complex distribution as shown in Figure 5(b), where $n = 3$ and the geometrical identity is $N_0 \cdot (N_1, s_1) \cdot (N_2, s_2)$. With different combinations of $N_k$ and $s_k$ ($k = 1, 2, ...$), $\mathcal{N}_A(i)$ can be catered to a known mismatch error distribution and yield a better performance. However, adding more component set(s) can increase the computational burden as the complexity increases rapidly with every additional set(s): $O(2^{N_0 + N_1 + N_2 + \cdots})$. Given mismatch error can be well rectified with a two-set implementation over a wide range of resolution, $n > 2$ might be unnecessary.

Similarly, three or more eyes may give better vision. However, the brain circuits and control network would become much more complicated to integrate signals and information. In fact, stereopsis is an advanced feature to human and animals with well-developed neural capacity [7]. Despite possessing two eyes, many reptiles, fishes and other mammals, have their eyes located on the opposite sides of the head, which limits the overlapping region thus stereopsis, in exchange for a wider field of vision. Certain species of insect such as *Arachnids* can possess from six to eight eyes. However, studies have pointed out that their eyes do not function in synchronous to resolve the fine resolution details [21]. It is not a coincidence that at least 30% of the human brain cortex is directly or indirectly involved in processing visual data [7]. We conjecture that the computational limitation is a major reason that many higher-order animals are evolved to have two eyes, thus keep the cyclops and triclops remain in the realm of mythology. No less as it would sacrifice visual processing precision, yet no more as it would overload the brain's circuit complexity.

## 4   Practical Implementation & Results

A mixed-signal ADC integrated circuit has been designed and fabricated to demonstrate the feasibility of the proposed architecture. The nature of hardware implementation limits the deployment of sophisticated learning algorithms. Instead, the circuit relies on a heuristic approach to efficiently estimate the mismatch error and adaptively reconfigure its components in an unsupervised manner. The detailed hardware algorithm and circuits implementation are presented seperately. In this paper, we only briefly summarize the techniques and results.

The ADC design is based on successive-approximation register (SAR) architecture and features redundant sensing with a geometrical identity $14 \cdot (13, 1)$. The component set $S_C$ is a binary-weighted capacitor array. We have chosen the smallest capacitance available in the CMOS process to implement the unit cell for reducing circuits power and area. However, it introduces large capacitor mismatch ratios up to 5% which limits the effective resolution to 10-bit or below for previous works reported in the literature [14, 19, 20].

The resolution of the secondary array is chosen as $N_1 = N_0 - 1$ to maximize the exchange capacity between two component sets

$$\bar{c}_{0,i} = \bar{c}_{1,i-1} = 1/2\bar{c}_{0,i+1}, \quad i \in [1, N - 2]. \tag{11}$$

In the auto-calibration mode, the mismatch error of each component is estimated by comparing the capacitors with similar nominal values implied by Equation (11). The procedure is unsupervised and fully automatic. The result is a reduced dimensional set of parameters that characterize the distribution of mismatch error. In the data conversion mode, a heuristic algorithm is employed that utilizes the estimated parameters to generate the component assembly with near-minimal mismatch error for each reference. A key technique is to shift the capacitor utilization towards the MSB by exchanging the components with similar weight, then to compensate the left-over error using the LSB. Although the algorithm has the complexity of $O(N_0 + N_1)$, parallel implementation allows the computation to finish within a single clock cycle.

By assuming the LSB components contribute an insignificant level of mismatch error as implied by Equation (5), this heuristic approach trades accuracy for speed. However, the excessive amount of

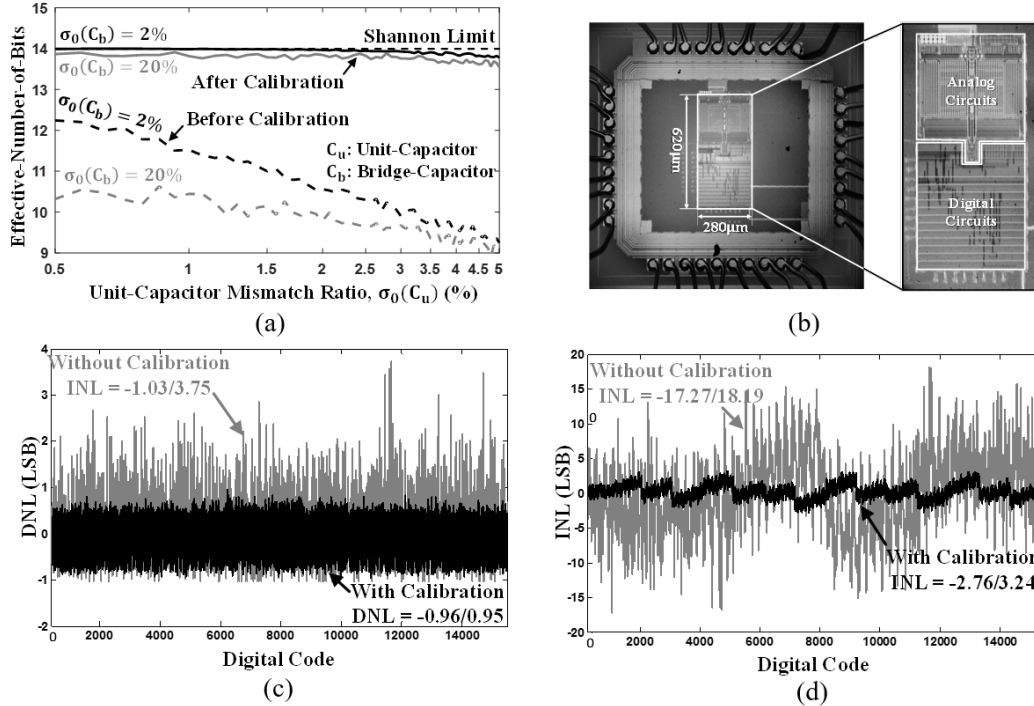

Figure 6: High-resolution ADC implementation. (a) Monte Carlo simulations of the unsupervised error estimation and calibration technique. (b) The chip micrograph. (c) Differential nonlinearity (DNL) and (d) integral nonlinearity (INL) measurement results.

redundancy guarantees the convergence of an adequate near-optimal solution. Figure 6(a) shows simulated plots of effective-number-of-bits (ENOB) versus unit-capacitor mismatch ratio, $\sigma_0(C_u)$. With the proposed method, the effective resolution is shown to approach the Shannon limit even with large mismatch ratios. It is worth mentioning that we also take the mismatch error associated with the bridge-capacitor, $\sigma_0(C_b)$, into consideration. Figure 6(b) shows the chip micrograph. Figure 6(c, d) gives the measurement results of standard ADC performance merit in terms of differential nonlinearity (DNL) and integral nonlinearity (INL). The results demonstrate that a 4-6 fold increase of linearity is feasible.

## 5    Conclusion

This work presents a redundant sensing architecture inspired by the binocular structure of the human visual system. We show architectural advantages of using redundant sensing in removing mismatch error and enhancing sensing precision. A high resolution, zero-dimensional data quantizer is presented as a proof-of-concept demonstration. Through Monte Carlo simulation with the error probabilistic distribution as a priori, we find the precision can approach the Shannon limit. In actual measurements without knowing the error probabilistic distribution, the gain of extra 2-bit precision and 4-6 times linearity is observed. We envision that the framework can be generalized to handle higher dimensional data and apply to a variety of applications such as digital imaging, functional magnetic resonance imaging (fMRI), 3D data acquisition, etc. Moreover, engineering such bio-inspired artificial systems may help better understand the biological processes such as stereopsis, microsaccade, and hyperacuity.

## Acknowledgment

The authors would like to thank Phan Minh Nguyen for his valuable comments.

## Footnotes

[1]$\theta_{2^N} = 2^N$ is a dummy reference to define the conversion full-range.

[2]The dummy reference $\theta_{2^N} = 2^N$ is exempted. Other references are normalized over the total weight to define the conversion full-range of FR $= [0, 2^N]$

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
