[Supplementary Material · nips_supplementary.pdf]

# A Bio-inspired Redundant Sensing Architecture (Supplementary File)

**Anh Tuan Nguyen, Jian Xu and Zhi Yang**[*]
Department of Biomedical Engineering
University of Minnesota
Minneapolis, MN 55455
[*]`yang5029@umn.edu`

This supplementary file presents the detailed implementation of a high-resolution analog-to-digital converter (ADC) in integrated circuits as mentioned in Section 4 "Practical Implementation & Results"

## 1 Component Sets

The ADC design utilizes a special case of the proposed redundant sensing architecture: $N = 14 = N_0 = N_1 + 1$ and $s_1 = 1$, which results in two component sets with nominal weights as follows

$$\begin{aligned}
\overline{S}_{C,0} &= \{1, 1, 2, 4, ..., 2^{N-2}\} \\
\overline{S}_{C,1} &= \{\quad 1, 2, 4, ..., 2^{N-2}\}.
\end{aligned} \tag{1}$$

The components are implemented using unit capacitors with the mean capacitance of 22fF and the effective mismatch ratio of 2-3%. We refer to the design as the "half-split" capacitor array since it is equivalent to dividing each conventional binary-weighted component into two identical halves

$$\begin{aligned}
\overline{c}_{0,i} + \overline{c}_{1,i-1} &= 2^i = \overline{c}_{0,i+1} \\
\overline{c}_{0,i} = \overline{c}_{1,i-1} &= 1/2\overline{c}_{0,i+1},
\end{aligned} \tag{2}$$

where $i \in [1, N-2]$. The relative mismatch error of each component $\varepsilon_{0,i}$ and $\varepsilon_{1,j}$ is defined as

$$\varepsilon_{i,j} = c_{i,j} - \overline{c}_{i,j} \frac{\sum_{\forall k,l} c_{k,l}}{2^N - 1}, \tag{3}$$

where $i \in [0, 1], j \in [0, N-1-i]$. By definition, the sum of all relative mismatch errors is equal to zero

$$\sum_{\forall i,j} \varepsilon_{i,j} = \sum_{\forall i,j} c_{i,j} - \sum_{\forall i,j} \overline{c}_{i,j} \cdot \frac{\sum_{\forall i,j} c_{i,j}}{2^N - 1} = 0. \tag{4}$$

## 2 SAR ADC Architecture

The functional block diagram of the ADC implementation is presented in Figure 1. The device has two "half-split" capacitor arrays in the differential configuration, a comparator, an on-chip memory and digital logic blocks. Each capacitor array is divided into three sections: MSB, LSB, and sub-DAC using the split-capacitor technique to reduce the total capacitance. The sub-DAC section is only used to fine-tune the error estimation procedure and represents $c_{0,0}$ during normal conversion. The primary logic blocks are composed of the conventional successive-approximation register (SAR) logic and the proposed capacitor mismatches estimation and calibration logic.

The circuit operations are comprised of two phases: the foreground error estimation and the normal conversion with background calibration. The error estimation procedures are engaged when the

Figure 1: Functional block diagram of the SAR ADC implementation.

device is reset, in which capacitor mismatches are automatically obtained and stored in the on-chip memory. The error estimation procedures only need to be performed once and take less than 10ms at 40kHz sampling rate. During the succeeding normal conversion phase, the calibration logic utilizes the estimated mismatches to generate the near-optimal component assemblies in real-time and adaptive to each input signal.

## 3  Capacitor Mismatches Estimation

The capacitor mismatches estimation is a mixed-signal process that automatically obtains the mismatches of all components. The key technique involves comparing and resolving the weight differences among consecutive components using charge-redistribution procedures. Because the mismatch error increases exponentially from LSB to MSB, the smallest LSB component, i.e. $c_{0,0}$, can be used as the reference to obtain the relative mismatches of the others.

The process is carried out in multiple iterations, each produces the mismatch of a component starting from LSB to MSB. At the beginning of an iteration, one of the differential capacitor arrays generates a voltage to counteract the comparator offset, while the capacitor comparison is performed on the other array. The ADC actually operates in the single-ended configuration during error estimation. The mismatches are obtained by evaluating the difference $(d_{0,i}, d_{1,i})$ between $(c_{0,i}, c_{1,i-1})$ and their preceding bit-capacitors $c_{0,i-1} + c_{1,i-2}$

$$
\begin{aligned}
d_{0,i} &= c_{0,i} - (c_{0,i-1} + c_{1,i-2}) \\
d_{1,i-1} &= c_{1,i-1} - (c_{0,i-1} + c_{1,i-2}),
\end{aligned}
\tag{5}
$$

where $i \in [1, N-1]$. Due to the relationship between components specified in Equation 2, $(d_{0,i}, d_{1,i})$ should have a zero-mean and a standard deviation proportional to the mismatch error.

The charge redistribution process is described in Figure 2 where $V_{ref+}$, $V_{ref-}$ and $V_{cm}$ represent the positive, negative and common-mode reference voltage respectively. The voltage $\Delta V_{\text{DAC}}$ generated on the top-plate is proportional to the difference

$$
\Delta V_{DAC} = d_{0,i} \frac{V_{ref+} - V_{ref-}}{\sum_{\forall j,k} c_{j,k}}.
\tag{6}
$$

$\Delta V_{\text{DAC}}$ is digitized with the remaining components from $c_{0,0}$ to $(c_{0,i-2}, c_{1,i-3})$ by switching the bottom-plates to either $V_{ref+}$ or $V_{ref-}$. The 4-bit sub-DAC array is used for fine resolving of $d_{0,i}$ and $d_{1,i-1}$ with the maximum precision of $1/16$ LSB.

After all $d_{0,i}$ and $d_{1,i-1}$ are resolved, the pre-estimated mismatches $\varepsilon'_{0,i}$ and $\varepsilon'_{1,i-1}$ are obtained by integrating the differences in a recursive manner.

$$
\begin{aligned}
\varepsilon'_{0,i} &= (\varepsilon_{0,i-1} + \varepsilon_{1,i-2}) + d_{0,i} \\
\varepsilon'_{1,i-1} &= (\varepsilon_{0,i-1} + \varepsilon_{1,i-2}) + d_{1,i-1},
\end{aligned}
\tag{7}
$$

where $i \in [1, N-1]$. There is systematic bias in values of $\varepsilon'_{0,i}$ and $\varepsilon'_{1,i-1}$ due to the assumption that $\varepsilon_{0,0} \approx 0$. This bias can be corrected by exploiting the property specified in Equation 4. The final

estimated mismatches $\varepsilon_{0,i}$ and $\varepsilon_{1,i-1}$ are obtained as follows

$$\begin{aligned}
\varepsilon_{0,i} &= \varepsilon'_{0,i} &&- S \cdot 2^{-n+i-1} \\
\varepsilon_{1,i-1} &= \varepsilon'_{1,i-1} &&- S \cdot 2^{-n+i-1},
\end{aligned} \tag{8}$$

where $i \in [1, N-1]$ and

$$S = \sum_{j=0}^{N-1} (\varepsilon'_{0,j} + \varepsilon'_{1,j-1}). \tag{9}$$

The estimated mismatches are stored in the on-chip memory as 10-bit fix-point numbers with a 6-digit integer. The result is a reduced-dimensional set of parameters that fully characterizes the mismatch error of capacitor arrays while only occupies a memory space of 480-bit.

Figure 2: The mixed-signal process to estimate capacitor mismatches based on charge-redistribution procedures.

## 4  Capacitor Mismatches Calibration

Figure 3(a) illustrates the calibration logic which operates in the background during normal data conversion. It is a heuristic algorithm that searches for the near-optimal component assembly to generate the required reference having the minimal error. The algorithm is designed to be efficient and can be implemented in a parallel structure such that the component assembly is recalculated within a single clock cycle, i.e. $O(1)$.

Figure 3: (a) Illustration of the on-chip capacitor mismatches calibration algorithm. (b) "Mapping & shifting" logic effectively migrates the component assembly towards the MSBs.

The calibration algorithm consists of two phases: "mapping and shifting" and residual error compensation. The first phase maps the digital code $\overline{D_{n-1}...D_1 D_0}$ to the component assembly without

considering the mismatch error while utilizing as much $S_{C,0}$ as possible as illustrated in Figure 3(b). The algorithm effectively migrates the component utilization towards the MSBs, leaving the many idle components in $S_{C,1}$ which can be used to compensate the residual error in the later stage.

The second phase involves computing the residual mismatches $\varepsilon_{\mathrm{res}}$ using the estimated parameters stored in the on-chip memory

$$\varepsilon_{\mathrm{res}} = \sum_{i=0}^{N-1} D_{0,i} \cdot \varepsilon_{0,i} + \sum_{j=0}^{N-2} D_{1,j} \cdot \varepsilon_{1,j} - V_{CO}, \tag{10}$$

where $D_{0,i}, D_{1,j} \in \{0,1\}$ are the digital digits associated with components in $S_{C,0}$, $S_{C,1}$ and generated from the output of the first phase. It is worth noting that the algorithm also compensates the comparator offset $V_{CO}$. The binary representation of the residual mismatches $\varepsilon_{\mathrm{res}}$ is then directly mapped to an assembly of the idle component in $S_{C,1}$ and is effectively compensated. Because the value of $\varepsilon_{\mathrm{res}}$ cannot exceed tens of LSB, the first 6 LSB components $(c_{1,0}, ..., c_{1,5})$ are sufficient, which mismatches are small enough to be neglected. It should be mentioned that some LSB components in $S_{C,1}$ could be already utilized from the first phase, thus are unavailable for residual error compensation. However, these scenarios only occur at digital codes near $2^N$, in which the mismatch error is sufficiently small. The Monte Carlo simulation results presented in the main paper have indicated that the residual mismatches $\varepsilon_{\mathrm{res}}$ can be effectively compensated in almost all situations.