[Reviews · NeurIPS 2016]

Reviewer 1

Summary

The paper argues that greater robustness to sensor msimatch (due either to irregular pixels or non-0uniform quantization in ADCs) can be achieved by taking multiple samples (not very surprising per se) and argue that this is happening in the human eye and explains the higher-than predicted visual accuity. They analyse the errors (for the ADC 'zero-dimensional case') in simulation and show that the mechanism permits achieving close to the shannon limit. They show a picture of a circuit and talk about capacitors but it's not clear if they've made a circuit or done any analysis of it.

Qualitative Assessment

not very clearly explained. e.g. you suddenly start talking about capacitors without explaining how they come into the picture. plausible idea, though the biological inspiration is a little implausible to me [not an expert]- you imply that there is an area of the brain with individual pixel representations, whereas AIUI by the LGN the brain is operating on a compressed representation (e.g. hubel & wiesel cells), so there is no part of the brain with access to "pixel" level information from both eyes. The same effect might be achieved with a higher-level representation, but that is not what you're arguing for. An across-micro-saccade monocular integration seems more plausible to me.

Confidence in this Review

1-Less confident (might not have understood significant parts)


Reviewer 2

Summary

Authors introduce an interesting redundant sensing architecture approach inspired by the binocular structure of the human visual system. In particular, the proposal allows removing mismatch error and enhances sensing precision. The approach is well motivated and explained from mathematical, biological, and hardware point of views. In addition, a high-resolution zero-dimensional data quantizer is presented as a proof of concept demonstration, including suitable simulations against realistic conditions and real-world results from a chip implementation.

Qualitative Assessment

The paper is well written and it was a pleasure to read it. In addition, the results presented are convincing and the real-world implementation experiments provide an added value to the work. Following, some minor comments are listed: - To clarify the influence of the heuristic solver in equation (10), it would be interesting to provide some quantitive results that allow comparing a more elaborate solution against the straightforward one that the authors employed.

Confidence in this Review

2-Confident (read it all; understood it all reasonably well)


Reviewer 3

Summary

The paper proposes a biologically inspired approach to design of analog-to-digital converter that is robust to mismatch present in a physical systems. The motivation comes from a redundancy in human visual system. The authors first explain the effect of mismatch on error of quantizers and propose an alternative, robust way to generate references for a quantizer. In particular, they introduce a second set of components which can contribute towards a reference computation. Their simulation and practical results show that the proposed solution outperforms conventional quantinaziers in the presence of mismatch.

Qualitative Assessment

The paper has a very well written introduction and provides a nice motivation for use of redundancy in sensing. A few comments: - It is hard to judge the implementation of the mismatch robust quantiser as there is no detailed explanation and I was not able to follow the section on implementation. - As a suggestion, I think it is also important to consider the increase in complexity of the system due to redundant sensing. This is not explicitly analysed in the paper. Overall, the paper presents a very interesting ideas, however, the important technical details are missing. Authors also say that a detailed hardware implementation will be presented in another submission, which makes this paper less suitable for this conference.

Confidence in this Review

1-Less confident (might not have understood significant parts)


Reviewer 4

Summary

The authors introduce an architecture for introducing redundancy in the reference value sets used by various ADC techniques (e.g. SAR). These degrees of freedom potentially allow one to optimize the combinations of references used to generate a threshold/comparison-point so as to minimize the error from component mismatch. An ADC circuit has been fabricated to implement such techniques, and they are indeed found to reduce mismatch error.

Qualitative Assessment

Update in response to author's rebuttal: Many thanks to the authors for taking the time to consider my points and to respond to them. In general, I would say that I look forward to seeing a revised version of the paper that makes the adjustments the authors promise, but the promised changes are sufficiently significant that (sight unseen) I only feel comfortable bumping up the technical quality by a point. My detailed responses to the responses, in order: -I do understand it is difficult to discuss too much prior work in the redundancy-for-mismatch-reduction space given the 9 page limitation. Nonetheless there is a complete absence of this at present in the paper, which may even count as a fatal flaw. (As an aside, note that McNeill's (JSSC 2005) work does *not* increase either power consumption or chip area (therein lies the innovation), although the authors are absolutely correct in noting that off-chip calibration power/space is unaccounted for. This is the kind of "Related work" discussion I would have appreciated seeing in the paper.) -While it is reassuring that the authors are planning to introduce more implementation details, my interest is less in the detailed implementation details and more in better motivation for the algorithmic choices that are made. With that said, if the paper were to spend more time discussing their (significant) contributions on the hardware implementation end, what I see as flaws in the algorithmic presentation would be far more forgivable. -The language may be simple, but it is surprisingly difficult to extract the core algorithmic ideas from a reading of the paper. This could very well be a reading comprehension issue on my end, but I doubt it. -Understood and agreed on the purpose of the simulation/implementation, and glad to hear that the authors are considering contrasting to stronger baselines. -I appreciate the clarification re: eq (5). It does make more sense to me now. -The author's listing of their assumptions in their rebuttal is much appreciated, but it is very difficult to parse from the paper. A similarly rational motivation for the structure of their design (N0/N1/etc.), as they promise, would also help a great deal. -Scalar quantization is 1-dimensional quantization. I remain unclear on why the authors are referring to it as zero-dimensional quantization (a term I neither understand nor have heard before --- and a simple google search seems to confirm its lack of usage in the literature). Positives: -The fundamental idea explored by the authors --- the use of redundancy to reduce mismatch errors --- is a compelling one. I would add that they are far from the first to explore this fertile area; see, for instance, McNeill's work (JSSC 2005, “Split ADC” Architecture for Deterministic Digital Background Calibration of a 16-bit 1-MS/s ADC). Their approach in adjusting which components to utilize represents a significant improvement over post-processing calibration. As the authors note, there are numerous situations where information is simply beyond recovering in the latter case. -While I am generally suspicious of "neural" analogies, the authors do an admirable job of drawing inspiration from the human visual system while optimizing a clear practical objective. -The authors implement their proposed solution not only in software and through simulations, but by fabricating an actual circuit. This adds tremendously to the credibility of their results. With that said, the authors do not go into details about their hardware implementation or its validation, and the focus of my review is on their algorithmic work (the authors themselves note that the hardware implementation is to be described elsewhere). (bigger) areas for improvement: -Fundamentally, I do not feel that the justification and explanation of the authors' algorithmic contributions is up to the NIPS standard. More on this below. -A lesser concern is the virtual lack of comparison to alternatives/nontrivial-baselines, either in terms of mismatch error vs quantization error, or in terms of the power/area tradeoff the authors frequently make reference to. This makes it very difficult to objectively evaluate the authors' contributions. -The model for binary coding introduced in Sec 2.2 is extremely difficult to follow ---one has to read through multiple times just to understand the terminology. What is a unit cell, versus a reference, versus a component? I was only able to piece this together through a combination of domain knowledge and guesswork from the expressions. -Furthermore, there is a lack of motivation for the binary coding approach beyond its use in "artificial systems" (it's relevant for SAR etc., but the authors provide no references or examples for this). This is critical, as the entire space of optimizations the authors are concerned with exist within this paradigm. -The model for component noise (eq. (5)) is supplied without any justification, either in words or in citations. Why, for instance, does the variance scale with the mean? Why not the standard deviation? -The description of the scheme in Section 3 is extraordinarily painful to decode. I would feel genuinely uncomfortable seeing this manuscript published the way this is currently written. One has to read the section twice to even understand the basic architecture that is being used, and terminology is used without any introduction/definition (e.g. the use of \hat{c} versus c --- there may even be a typo on line 269 as the \hat is used in the argument but not the subscript, but I would not be able to swear by this, as the \hat significance is never provided.). -Related to the above, the form of the proposed solution (eq. (8)) is not sufficiently motivated or justified. A red flag: not only do the authors never explain the purpose/intuition behind parameters N_0/N_1/s_1, but they later turn to simulations (lines 310-313) just to explain their impact on the solution. If there are simple insights that lead one to the system designed by the authors, this manuscript certainly does not make them clear. -Results reported for DNL/INL make for fantastic confirmations that the techniques proposed work in practice. With that said, confirmation of functionality is insufficient in a space as crowded as this one (mismatch calibration/correction), and performance needs to be compared against nontrivial and informative baselines. (smaller) areas/points for improvement: -The authors frequently refer to scalar quantization as zero-dimensional. Unless I am missing something big, I believe they mean one-dimensional. -Eq. (6): should NOT index c_i by i, since that index is being used with a different interpretation by X_i. -line 213: should be fig. 3(a). -What is fig. 3a showing? Expected absolute error over many samples, or a single sample? -line 241: \in [0,1] should be \in \{0,1\}, as the former refers to the unit interval.

Confidence in this Review

2-Confident (read it all; understood it all reasonably well)


Reviewer 5

Summary

Explores a redundant sensing architecture taking as use case analog to digital conversion and inspired by overlapping field of vision in higher mammals, in particular human vision which can achieve down to 2-5 arc second resolution where the retinal hardware is limited to 20-30 arc seconds. Mismatch error of the quantized codes in the ADC is presented as a function of quantizer resolution, where it is seen that in higher resolution quantizers this type of error dominates over quantization error. Current approaches to minimize the mismatch error are focused on using larger components with increased cost and power. The proposed solution is to use two component sets N0 and N1, which results in 2^(N0+N1) possible assemblies much greater than the needed reference set of 2^N0. Differences in the components result from fabrication variations. Generating the reference value which minimizes mismatch error through optimal component assemblies is and NP-optimization problem so a heuristic approach is taken in the constructed ADC. Most of the mismatch error comes from the most significant bit so capacitor utilization is shifted toward the MSB. In practice an extra 2bit of precision is gained and the effective resolution approaches the Shannon limit.

Qualitative Assessment

Not having experience in IC design and fabrication what is the impact on chip area with supporting both primary and secondary component sets? Is it possible to approach the optimal component assemblies using a Viterbi algorithm, assigning components from MSB to LSB? It is commented in the introduction that redundancy is an important characteristic to neural networks. Do you mean redundancy in weights within the same network, or the case where separate networks are trained from the same input, but from a different initialization. The performance from ensembling the resulting decision from separate networks often results in performance gains over that achievable with single larger networks. Is there a connection to the redundancy you explore here.

Confidence in this Review

2-Confident (read it all; understood it all reasonably well)